# PGRMC1 Regulates Cellular Senescence via Modulating FOXO1 Expression in Decidualizing Endometrial Stromal Cells

**DOI:** 10.3390/biom12081046

**Published:** 2022-07-28

**Authors:** Atsuya Tsuru, Mikihiro Yoshie, Junya Kojima, Ryo Yonekawa, Mana Azumi, Kazuya Kusama, Hirotaka Nishi, Kazuhiro Tamura

**Affiliations:** 1Department of Endocrine Pharmacology, Tokyo University of Pharmacy and Life Sciences, Tokyo 192-0392, Japan; y151119@toyaku.ac.jp (A.T.); yonex2834@gmail.com (R.Y.); azumi@toyaku.ac.jp (M.A.); kusamak@toyaku.ac.jp (K.K.); hiro@toyaku.ac.jp (K.T.); 2Department of Obstetrics and Gynecology, Tokyo Medical University, Tokyo 160-0023, Japan; kojima_j@tokyo-med.ac.jp (J.K.); nishih@tokyo-med.ac.jp (H.N.)

**Keywords:** PGRMC1, decidualization, senescence

## Abstract

The appropriate differentiation of endometrial stromal cells (ESCs) into decidual cells is required for embryo implantation and subsequent placentation into humans. Decidualization is accompanied by the appearance of senescent-like cells. We recently reported the secretory phase-specific downregulation of endometrial progesterone receptor membrane component 1 (PGRMC1) and enhanced decidualization upon PGRMC1 knockdown and inhibition in cultured ESCs. However, it remains unknown whether PGRMC1 is involved in cellular senescence during decidualization. Here, we showed that the small interfering RNA (siRNA)-mediated knockdown of PGRMC1 and the inhibition of PGRMC1 by AG-205 increased the expression of the transcription factor forkhead box protein O1 (FOXO1) and the senescence-associated β-galactosidase activity in cAMP analog- and progesterone-treated ESCs. Furthermore, the knockdown of FOXO1 repressed the decidual senescence induced by siRNA-based PGRMC1 knockdown or AG-205 treatment. Taken together, the decreased PGRMC1 expression in ESCs may accelerate decidualization and cellular senescence via the upregulation of FOXO1 expression for appropriate endometrial remodeling and embryo implantation during the secretory phase.

## 1. Introduction

Pregnancy is dependent upon the success of various steps including ovulation, fertilization, blastocyst development, endometrial differentiation (decidualization), embryo implantation, and placental formation. Among them, the differentiation of endometrial stromal cells (ESCs) into decidual cells is essential for the acquisition of endometrial receptivity for implantation and subsequent placentation. During the menstrual cycle, the estrogen-primed proliferative endometrium is decidualized by the action of progesterone (P4) and cAMP in the mid-secretory phase. Decidualization is characterized by the transformation of ESCs into secretory decidual cells, which produce insulin-like growth factor-binding protein 1 (IGFBP1) and prolactin (PRL) [1,2,3,4]. The treatment of cultured ESCs with P4 and a cAMP analog in vitro can replicate the decidualization in vivo [1,5]. This in vitro decidualization model is widely used to elucidate the molecular mechanism of decidualization. Recent studies provide evidence that decidualization is accompanied by cellular senescence [6,7,8,9]. In general, DNA damage results in cellular senescence, which causes irreversible cell cycle arrest [10,11].

Senescent cells produce various secretory factors including inflammatory cytokines such as interleukins (ILs) and chemokines. This phenotype is called the senescence-associated secretory phenotype (SASP), and SASP factors further induce senescence of surrounding cells. In cancer cells, the SASP stimulates tumor growth and metastasis and triggers an immune recognition and tumor clearance [12]. Although many questions currently remain unanswered concerning the cancer development that is associated with cellular senescence, senescence-based therapies warrant further investigation [13].

It has been reported that a member of the forkhead family of transcription factors, forkhead box protein O1 (FOXO1), is associated with the expression of decidualization markers such as IGFBP1 and PRL [14]. FOXO1 may be involved in IL8-dependent ESC senescence [6]. The SASP factors IL6 and IL8 are secreted by ESCs. Senescent decidual cells may be eliminated by uterine-specific natural killer (NK) cells that are activated by the IL15 secreted by decidual cells [6]. In the process of decidualization, cell differentiation and senescence should be maintained at an appropriate balance, and failure to maintain this balance might be associated with abnormal decidualization [7,15] and preterm birth [8,9,16].

The post-ovulatory increase in P4 in the corpus luteum is involved in decidualization [17]. P4 mostly functions via the intracellular classical progesterone receptor (PGR). Once P4 binds to the PGR, the complex translocates into the nucleus and then binds directly to the DNA of a specific gene in order to regulate transcription. The importance of P4 action via the PGR is evident from the infertility of PGR-deficient mice and the association of abnormal P4 signals with diseases such as endometriosis [18] and breast cancer [19,20], miscarriage, and preterm birth [9]. Progesterone receptor membrane component 1 (PGRMC1) is a non-classical P4 receptor that differs from the classical PGR [21]. It forms a dimer to exert its action. PGRMC1 dimerizes via heme and the dimer interacts with cytochrome P450 and epidermal growth factor receptor (EGFR) in order to increase cancer cell growth and resistance to chemotherapy [22]. In reproductive organs, PGRMC1-mediated P4 signaling suppresses the apoptosis of human ovarian granulosa cells [23], and uterine-specific PGRMC1-deficient mice exhibit a reduced fertility and form cysts in the endometrium [24]. In triple negative breast cancer cells, the negative effect of P4 on cell proliferation and cell death was reduced by repressing PGRMC1 [25].

We and others previously demonstrated that endometrial PGRMC1 expression decreases during the secretory phase of the menstrual cycle and the small interfering RNA (siRNA)-mediated knockdown and inhibition of PGRMC1 promotes ESC decidualization in vitro [26,27]. However, the relationship between the PGRMC1 expression and cellular senescence in ESCs during decidualization has not been explored. Therefore, the present study investigated the role of PGRMC1 in cellular senescence accompanying decidualization.

## 2. Materials and Methods

### 2.1. Reagents and Antibodies

P4 (#P0130), dibutyryl-cAMP (db-cAMP, #D0260), and AG-205 (#A1487) were purchased from Sigma-Aldrich (St. Louis, MO, USA). Antibodies against PGRMC1 (D6M5M) and FOXO1 (C29H4) were obtained from Cell Signaling Technology (Danvers, MA, USA). The antibody against glyceraldehyde-3-phosphate dehydrogenase (GAPDH, clone 5A12) was from FUJIFILM Wako Pure Chemical Corporation (Osaka, Japan).

### 2.2. Cell Culture

Human endometrial tissue samples were collected from female patients with leiomyoma and who were undergoing surgery at the Tokyo Medical University Hospital. All patients (*n* = 3) were younger than 45 years and had regular 28–32 day menstrual cycles. Endometrial tissues in the proliferative phase were prepared for ESC culture as described previously [5]. Briefly, the endometrial tissue was digested with 0.25% collagenase (Type IA, Sigma-Aldrich) and then strained through 250 and 38 μm stainless steel sieves in order to remove undigested tissue and glandular cells, respectively. At least 95% of isolated cells were positive for vimentin, but these cells were not stained for cytokeratin. The ESCs were maintained in phenol red-free Dulbecco’s Modified Eagle’s medium/F12 medium (Life Technologies, Tokyo, Japan) supplemented with 10% (*v*/*v*) charcoal-treated fetal bovine serum (HyClone, South Logan, UT, USA), 50 U/mL penicillin, 50 mg/mL streptomycin, and 0.25 mg/mL fungizone (Life Technologies). In order to induce decidualization, the ESCs were treated for two days with db-cAMP (0.5 mM) and P4 (1 μM), abbreviated as D/P.

### 2.3. siRNA Transfection

ESCs grown to 50% confluency in 24-well plates were transfected with non-targeting control siRNA (30 pmol; Qiagen, Mississauga, ON, Canada), PGRMC1 siRNA (30 pmol; EHU003641, Sigma-Aldrich) or FOXO1 siRNA (30 pmol, Sigma-Aldrich), using Lipofectamine RNAiMAX Transfection Reagent (Invitrogen, Carlsbad, CA, USA) according to the manufacturer’s instructions [28]. After treatment with siRNA, the cells were washed with a phosphate-buffered saline (PBS) and treated with a medium containing or lacking P4 or cAMP analogs for 2 days.

### 2.4. Immunoblotting

Protein lysates were prepared from ESCs with a RIPA buffer (Nakalai Tesque Inc., Kyoto, Japan) according to the manufacturer’s instructions. Equal amounts of protein (20 μg) were separated by SDS-PAGE and electrophoretically transferred into polyvinylidene difluoride membranes. The membranes were probed with primary antibodies against PGRMC1 (1:2500) and FOXO1 (1:2500). Immunoreactive bands were detected using enhanced chemiluminescence after incubation with a horseradish peroxidase-labeled anti-mouse or anti-rabbit IgG. The membranes were then treated with a stripping solution and re-probed with an antibody against GAPDH (1:10,000). The relative band intensity was evaluated by the densitometric analysis of digitalized autographic images using ImageJ software (NIH, Bethesda, MD, USA) and normalized to that of GAPDH [28].

### 2.5. RNA Extraction and Real-Time RT-PCR

The total RNA was extracted using the reagent ISOGEN II (Nippon Gene, Tokyo, Japan) according to the manufacturer’s instructions. The RNA (100 ng) was amplified by real-time RT-PCR using a SYBR Green Luna Universal One-step RT-qPCR Kit (New England Biolabs, Beverly, MA, USA). Sense (S) and antisense (AS) primers were 5′-AAGGGTGACAGCAACAGCTC-3′ (S) and 5′-TTCTGCACACGAATGAACTTG-3′ (AS) for PGRMC1, 5′-AAGGGTGACAGCAACAGCTC-3′ (S) and 5′-TTCTGCACACGAATGAACTTG-3′ (AS) for FOXO1, 5′-AGACAGCCACTCACCTCTTCAG-3′ (S) and 5′-TTCTGCCAGTGCCTCTTTGCTG-3′ (AS) for IL6, 5′-AAGACATACTCCAAACCTTTCCA-3′ (S) and 5′-CCAGACAGAGCTCTCTTCCA-3′ (AS) for IL8, 5′-AACAGAAGCCAACTGGGTGAATG-3′ (S) and 5′-CTCCAAGAGAAAGCACTTCATTGC-3′ (AS) for IL15, and 5′-AGCCACATCGCTCAGACA-3′ (S) and 5′-GCCCAATACGACCAAATCC-3′ (AS) for GAPDH. The fold change in the expression of each gene was calculated using the ΔΔCt method with GAPDH as an internal control [28].

### 2.6. Senescence-Associated β-Galactosidase (SA-β-Gal) Staining

SA-β-Gal staining was performed as previously described [7]. Briefly, ESCs were treated with AG-205; a PGRMC1 inhibitor (10 μM), PGRMC1 siRNA, and/or D/P were fixed in 4% (*w*/*v*) paraformaldehyde for 10 min and then washed twice with PBS. The staining was performed overnight at 37 °C in PBS (pH 6.0) containing 5 mM potassium ferrocyanide, 5 mM potassium ferricyanide, 1 mM MgCl_2_, and 1 mg/mL X-gal. The fold change in SA-β-Gal-positive cells was evaluated in five randomly chosen fields and expressed and compared with the control group.

### 2.7. Cell Cycle Analysis by Flow Cytometry

ESCs treated with AG205 or PGRMC1 siRNA were incubated with D/P for 2 days and detached with trypsin-EDTA. The cells were resuspended in the medium containing Hoechst 33342 (0.01 mg/mL, Dojindo, Kumamoto, Japan) and incubated for 45 min at 37 °C in the dark. The cell suspension was analyzed by flow cytometry (SH800 Cell Sorter, Sony Biotechnology Inc., Tokyo, Japan). The percentage of cells in the different phases of the cell cycle was calculated with the FlowJo software (ver.10.8.1, BD Biosciences, Ashland, OR, USA).

### 2.8. Enzyme-Linked Immunosorbent Assay (ELISA)

The culture medium was centrifuged at 10,000× *g* at 4 °C for 10 min, and the supernatant was collected in order to determine the IL8 concentration. The concentrations of IL6 and IL8 were determined using an ELISA kit (Abcam ab178013 Human IL-6 SimpleStep ELISA kit and LBIS Human IL8 (CXCL8) ELISA Kit) according to the kit instructions, respectively.

### 2.9. Statistical Analysis

The data are expressed as the mean ± standard error of the mean (SEM) from at least three independent experiments. The significance was assessed using the Dunnett multiple comparisons test. A *p*-value of less than 0.05 was considered statistically significant.

## 3. Results

### 3.1. PGRMC1 Inhibition and Knockdown Induce Cellular Senescence in Decidualizing ESCs

In order to examine the role of PGRMC1 in cellular senescence in decidualizing ESCs, we evaluated the effects of a PGRMC1 inhibitor, AG-205, that blocks PGRMC1 action [29,30,31], and the knockdown of PGRMC1 on SA-β-Gal activity in db-cAMP and P4 (D/P)-treated decidualizing cells (Figure 1). The ESCs were pre-treated with AG-205 or siRNA and then stimulated for 2 days with D/P. The SA-β-Gal activity increased in D/P-treated decidualizing cells (Figure 1A,B). AG-205 treatment increased the number of SA-β-Gal-positive cells compared with the number of control cells (Figure 1A,B). Treatment with AG-205 enhanced SA-β-Gal activity in the D/P-stimulated decidualizing cells (Figure 1A,B). The transfection of PGRMC1-specific siRNA significantly reduced its mRNA and protein expression in ESCs (Figure 1C,D) but increased the intensity of SA-β-Gal staining. The D/P-induced SA-β-Gal staining tended to increase in the PGRMC1-knockdown cells (Figure 1E,F). The cell cycle analysis showed that treatment with AG205 or PGRMC1 siRNA increased the population during the G2/M phase (Figure 1G). These results suggest that PGRMC1 is possibly involved in the senescence of ESCs.

### 3.2. PGRMC1 Inhibition and Knockdown Alter the Expression of Senescence-Associated Inflammatory Factors in ESCs

We next evaluated the effects of PGRMC1 inhibition and silencing on the expression of senescence-associated inflammatory factors in ESCs. Although *IL8* mRNA was downregulated in the D/P-stimulated decidualizing cells (Figure 2A), the IL8 secretion was increased by D/P treatment (Figure 2B). Treatment with AG-205 further increased IL8 secretion in the D/P-treated cells (Figure 2B). In contrast, PGRMC1 siRNA decreased IL8 secretion in the D/P-stimulated cells (Figure 2B).

*IL6* expression was suppressed in the D/P-stimulated ESCs (Figure 2C). AG205 increased mRNA expression and the secretion of IL6 in the presence of D/P. However, PGRMC1 knockdown did not alter *IL6* mRNA expression and the secretion under D/P stimulation (Figure 2C,D). Decidualization increases the secretion of IL15. Treatment with the PGRMC1 inhibitor or siRNA did not alter the mRNA expression of *IL15* in the absence of the decidual stimulus. Treatment with AG-205 plus D/P reduced *IL15* expression compared with the D/P treatment only. PGRMC1 siRNA increased the expression of *IL15* in decidualized cells in the presence of D/P (Figure 2E).

### 3.3. PGRMC1 Inhibition and Knockdown Promote the Expression of D/P-Induced FOXO1 in ESCs

FOXO1 is crucial for the induction of decidual markers [14] and is involved in decidual senescence [6] in ESCs. Therefore, the effects of PGRMC1 inhibition and knockdown on FOXO1 expression were examined. The D/P treatment upregulated FOXO1 expression (Figure 3A,B). Although treatment with AG-205 alone did not alter FOXO1 expression, the D/P-induced FOXO1 expression was further increased by AG-205 (Figure 3A). Furthermore, the D/P-stimulated FOXO1 expression was enhanced in PGRMC1-depleted cells (Figure 3B).

### 3.4. FOXO1 Knockdown Suppresses Decidual Senescence in ESCs

FOXO1 knockdown suppresses senescence [6]. Two types of FOXO1 siRNA were tested in order to evaluate the effect on the expression of FOXO1. Each FOXO1 siRNA greatly suppressed FOXO1 expression (Figure 4A). FOXO1 knockdown suppressed the SA-β-Gal staining accompanying decidualization in ESCs (Figure 4B,C).

### 3.5. FOXO1 Silencing Represses Decidual Senescence Induced by PGRMC1 Inhibition and Knockdown

To elucidate the relationship between PGRMC1 and FOXO1 in decidual senescence, we examined the effects of FOXO1 knockdown on decidual senescence upon PGRMC1 inhibition and knockdown (Figure 5). The double knockdown of PGRMC1 and FOXO1 in decidualized ESCs was confirmed by immunoblotting (Figure 5A). The enhanced SA-β-Gal activities in ESCs treated with AG-205 and D/P were completely repressed by FOXO1 knockdown (Figure 5B,C). In addition, FOXO1 silencing suppressed the elevation of SA-β-Gal-positive cells upon treatment with PGRMC1 siRNA and D/P (Figure 5D,E). This study suggests that FOXO1 was involved in the mechanism of accelerated cellular senescence due to decreased PGRMC1 expression associated with decidualization (Figure 6).

## 4. Discussion

In the human endometrium, PGRMC1 expression is lowered during the secretory phase of the menstrual cycle [26,32]. We have shown that the PGRMC1 level is downregulated during decidualization of cultured ESCs in vitro and that a microRNA, miR-98, is involved in PGRMC1 downregulation in ESCs [27]. Further, both PGRMC1 inhibition by AG-205 and knockdown by siRNA promote the expression of the decidual markers IGFBP1 and PRL in ESCs [27]. Salsano et al. [26] reported that the overexpression of PGRMC1 in ESCs abrogates decidualization. Thus, the downregulation of PGRMC1 may promote decidualization during the secretory phase in the endometrium for the establishment of pregnancy.

Decidualization is accompanied by the appearance of senescent cells [6,7,8]. Cellular senescence is accompanied by the persistent arrest of cell proliferation during the process of differentiation. Decidual senescence is a critical component of decidualization and necessary for the initial pro-inflammatory response required for embryo implantation. Our group demonstrated that senescent cell removal by senolytic agents stimulates decidualization [33]. However, it remains unknown whether PGRMC1 is involved in decidual senescence. In the present study, we demonstrated that the knockdown and inhibition of PGRMC1 increased SA-β-Gal activity, an indicator of cellular senescence, and the population of G2/M phase in decidualizing ESCs.

Although the precise mechanism of decidual senescence has not been well characterized, Brighton et al. reported that FOXO1, which is upregulated upon decidualization, functions as a key inducer of cellular senescence via upregulating IL8 in ESCs [6].

In this study, PGRMC1 knockdown and inhibition stimulated FOXO1 expression in the D/P-treated ESCs. Of note, FOXO1 knockdown suppressed the cellular senescence induced by PGRMC1 knockdown and inhibition. These results suggest that the downregulation of PGRMC1 in ESCs may cause cellular senescence by increasing FOXO1 expression upon decidualization (Figure 6). In addition, cellular senescence was observed in the PGRMC1 inhibitor- and siRNA-treated cells even in the absence of the decidual stimulus. This suggests that PGRMC1 also suppresses cellular senescence in a FOXO1-independent manner in undifferentiated ESCs. Interestingly, age-related endometrial cysts in the uterus form earlier in PGRMC1-knockout mice than in normal mice [34]. The formation of endometrial cysts is a typical phenotype of senescence in mice and humans, suggesting that there is a link between PGRMC1 and senescence in the uterus. In the fetal membrane, abnormal senescence can cause preterm birth, preeclampsia, and intrauterine growth restrictions [35]. Oxidative stress, characterized by imbalances in the redox system in the fetal–maternal intrauterine compartments, has been reported to play a crucial role in the pathogenesis of preeclampsia [36,37] and preterm birth [38]. Hydrogen peroxide-triggered senescence is further promoted by the knockdown of PGRMC1 in fetal membrane cells [35], suggesting that PGRMC1 protects against oxidative stress-induced senescence in the fetal membrane.

As SASP factors, IL8 and IL6 protein secretion increased with PGRMC1 inhibition, whereas *IL8* mRNA expression was not altered. The inflammatory reaction was initiated in the early stage of decidualization [39]. We confirmed that the expression of *IL8* mRNA was promoted at 24 h after decidual stimulation, and the AG-205 treatment increased the expression of *IL8* (data not shown). These results imply that *IL8* mRNA expression may be increased by PGRMC1 inhibition and knockdown at an earlier timepoint than analyzed in this study. Similarly for another SASP factor, IGFBP7, our preliminary data showed that the expression was increased by the decidualization stimuli and that the treatment of AG205 promoted the D/P-stimulated IGFBP7 expression (data not shown). Furthermore, the expression of p21 and p53 were not changed by AG205 and PGRMC1 knockdown in ESCs (data not shown). These results suggest that enhanced cellular senescence by PGRMC1 inhibition and knockdown may not be mediated by p53 and p21 induction in ESCs.

The IL15 secreted by decidualized cells activates the NK cells in the uterus and induces the elimination of senescent cells by activating the NK cells [6]. Although the expression of *IL15* mRNA increased upon decidualization, different responses were detected upon PGRMC1 inhibition and knockdown. This result may be due to the difference in culture time between experiments using AG-205 and PGRMC1 siRNA. PGRMC1 exhibits haem-dependent dimerization on the cell membrane [22]. The PGRMC1 dimer binds to EGFR [40] and cytochrome P450 [40,41] and thereby enhances tumor cell proliferation and chemotherapy resistance [42]. AG-205 may inhibit the dimerization of PGRMC1, and the dissimilarity between inhibition and silencing of PGRMC1 might result in different outcomes. AG-205 has been widely used as an PGRMC1 inhibitor; however, it has some PGRMC1-independent inhibitory effects on the galactosylceramide synthesis and stimulatory effects on large vesicular structure formations, cholesterol biosynthesis, and steroidogenesis [42,43,44,45]. Although we cannot exclude the possibility that a non-specific action of AG-205 affects the expression of the SASP factors, the effects of AG-205 on decidualization and FOXO1-mediated decidual senescence were similar to those of PGRMC1 knockdown. This issue requires further investigation in order to clarify the role of PGRMC1 in ESCs.

Previously, we reported that PGRMC1 inhibition and knockdown enhanced db-cAMP-induced decidualization, even in the absence of P4. We therefore speculated that the action of P4 was not necessary for the stimulatory effect of PGRMC1 downregulation on decidualization. The expression of FOXO1 was increased by the activation of cAMP signaling in decidualizing ESCs [46]. Although the precise mechanism by which PGRMC1 knockdown and inhibition induce FOXO1 in decidualizing ESCs is unclear, PGRMC1 might regulate cAMP signaling in ESCs. Thus, it is necessary to clarify the relationship between cAMP signaling and PGRMC1 in decidualizing ESCs. The elucidation of the regulatory action of PGRMC1-mediated FOXO1 expression may improve our understanding of the molecular mechanisms underlying differentiation and cellular senescence during ESC decidualization.

A limitation of this study is that we evaluated PGRMC1 in cultured ESCs but not in the endometrium in vivo. Human endometrial assembloids consisting of gland-like organoids and primary stromal cells were recently established [45]. They can respond to decidual stimuli and resemble the mid-luteal endometrium that has differentiated and senescent subpopulations of endometrial cells [45]. The use of this model may further clarify the precise role of PGRMC1 in endometrial decidualization and accompanying senescence.

In the present study, the functional inhibition of PGRMC1 promoted not only decidualization but also decidual senescence in ESCs (Figure 6). In conclusion, our data suggest that PGRMC1 in the endometrium suppresses cellular senescence and that the secretory phase-specific downregulation of PGRMC1 expression induces appropriate decidualization in order to enable remodeling of the endometrium and embryo implantation.

## Figures and Tables

**Figure 1 biomolecules-12-01046-f001:**
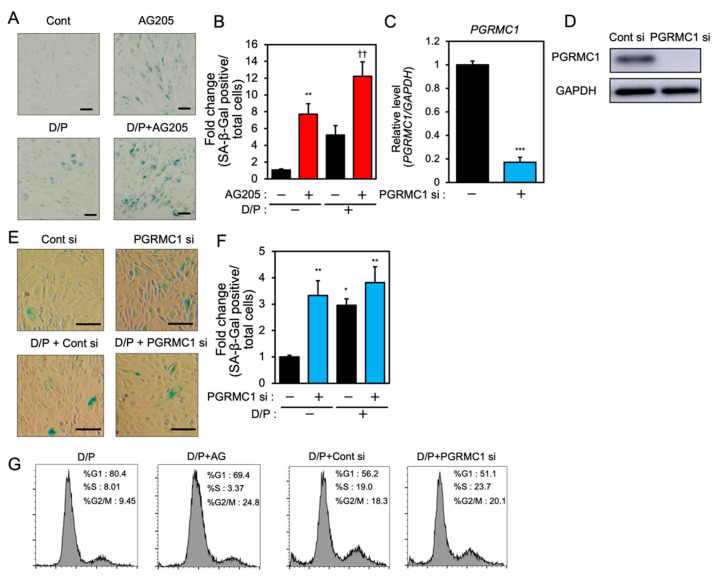
PGRMC1 inhibition and knockdown induce cellular senescence in ESCs. (**A**) ESCs were pre-treated with the PGRMC1 inhibitor AG-205 (10 μM) for 1 h and then stimulated with db-cAMP (500 μM) and P4 (1 μM) (D/P) for 2 days. The ESCs were stained for SA-β-Gal. (**B**) The graph shows the relative fold change in SA-β-Gal-positive cells. Red bars show the AG205-treated group. Values represent mean ± SEM from three independent experiments with different ESCs. ** *p* < 0.01 vs. control. †† *p* < 0.01 vs. D/P. (**C**) ESCs were transfected with a control or PGRMC1 siRNA for 1 day and treated with D/P for 2 days. The PGRMC1 mRNA expression was evaluated by real-time RT-PCR. Blue bar shows the PGRMC1 siRNA-transfected group. *** *p* < 0.001 vs. control. (**D**) The PGRMC1 protein level was evaluated by immunoblotting. (**E**) Cells were stained for SA-β-Gal. (**F**) The graph shows the relative fold change in SA-β-Gal-positive cells. Values represent mean ± SEM from three independent experiments. Blue bars show the PGRMC1 siRNA-transfected group. * *p* < 0.05, ** *p* < 0.01 vs. control. The bar in the figure (**A**,**E**) represents 100 μm. (**G**) ESCs treated with AG205 or PGRMC1 siRNA were stimulated with D/P for 2 days and analyzed by flow cytometry.

**Figure 2 biomolecules-12-01046-f002:**
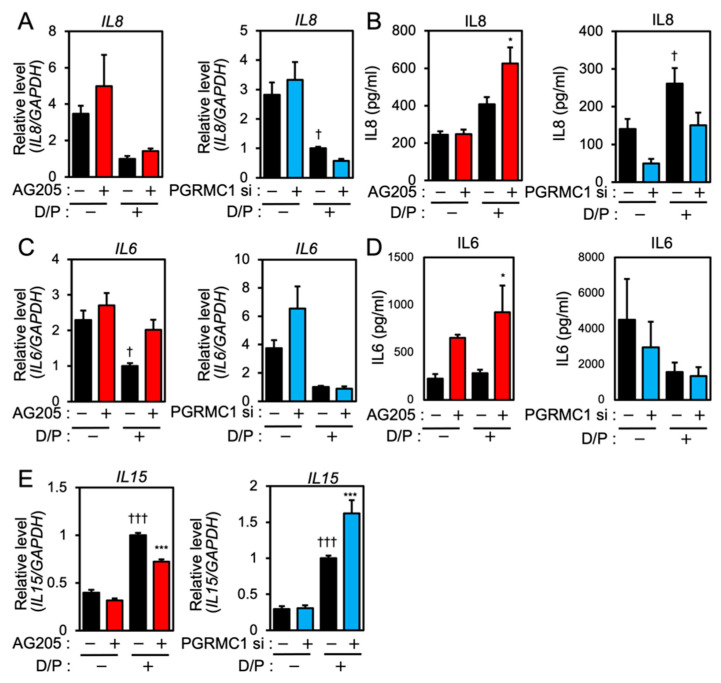
PGRMC1 inhibition and knockdown alter the expression of SASP-related ILs in ESCs. ESCs were pre-treated with the PGRMC1 inhibitor AG-205 (10 μM) for 1 h or PGRMC1 siRNA for 1 day and then stimulated with D/P for 2 days. *IL8* (**A**), *IL6* (**C**), and *IL15* (**E**) mRNA levels were determined by real-time RT-PCR. GAPDH was used as the reference gene. † *p* < 0.05 vs. control. ††† *p* < 0.001 vs. control. *** *p* < 0.001 vs. D/P. IL8 (**B**) and IL6 (**D**) secreted into the cell culture medium were analyzed with ELISA. Red and blue bars show the AG205-treated and PGRMC1 siRNA-transfected group, respectively. * *p* < 0.05 vs. D/P.

**Figure 3 biomolecules-12-01046-f003:**
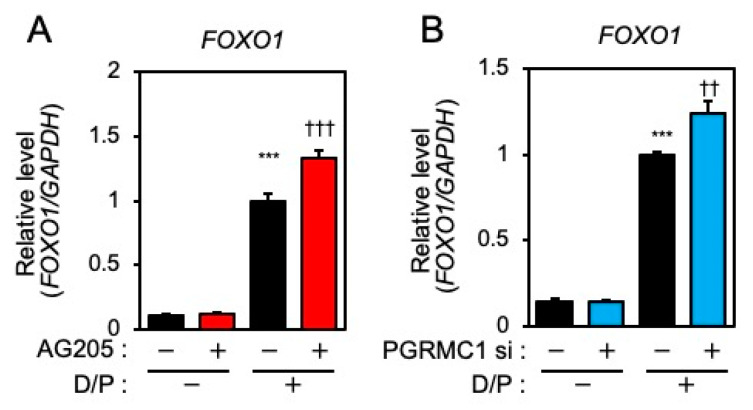
PGRMC1 inhibition and knockdown promote D/P-induced FOXO1 expression. ESCs were pre-treated with the PGRMC1 inhibitor AG-205 (**A**) for 1 h or PGRMC1 siRNA (**B**) for 1 day and then stimulated with D/P for 2 days. Expression of FOXO1 was determined by real-time RT-PCR. GAPDH was used as the reference gene. Red and blue bars show the AG205-treated and PGRMC1 siRNA-transfected group, respectively. *** *p* < 0.001 vs. control. †† *p* < 0.01, ††† *p* < 0.001 vs. D/P.

**Figure 4 biomolecules-12-01046-f004:**
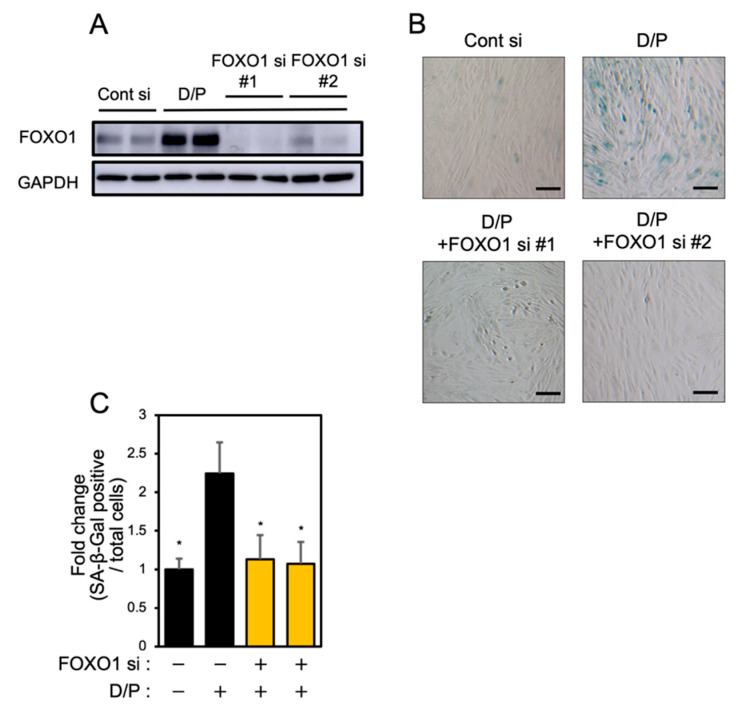
FOXO1 knockdown suppresses decidual senescence in ESCs. ESCs were transfected with FOXO1 siRNA for 1 day and then stimulated with D/P for 2 days. (**A**) FOXO1 expression was determined by immunoblotting analysis. (**B**) ESCs were stained for SA-β-Gal. Scale bars = 100 μm. (**C**) The graph shows the relative ratio of SA-β-Gal-positive cells to total cells. Yellow bars show FOXO1 siRNA transfected groups. Values represent mean ± SEM from three independent experiments. * *p* < 0.05 vs. D/P.

**Figure 5 biomolecules-12-01046-f005:**
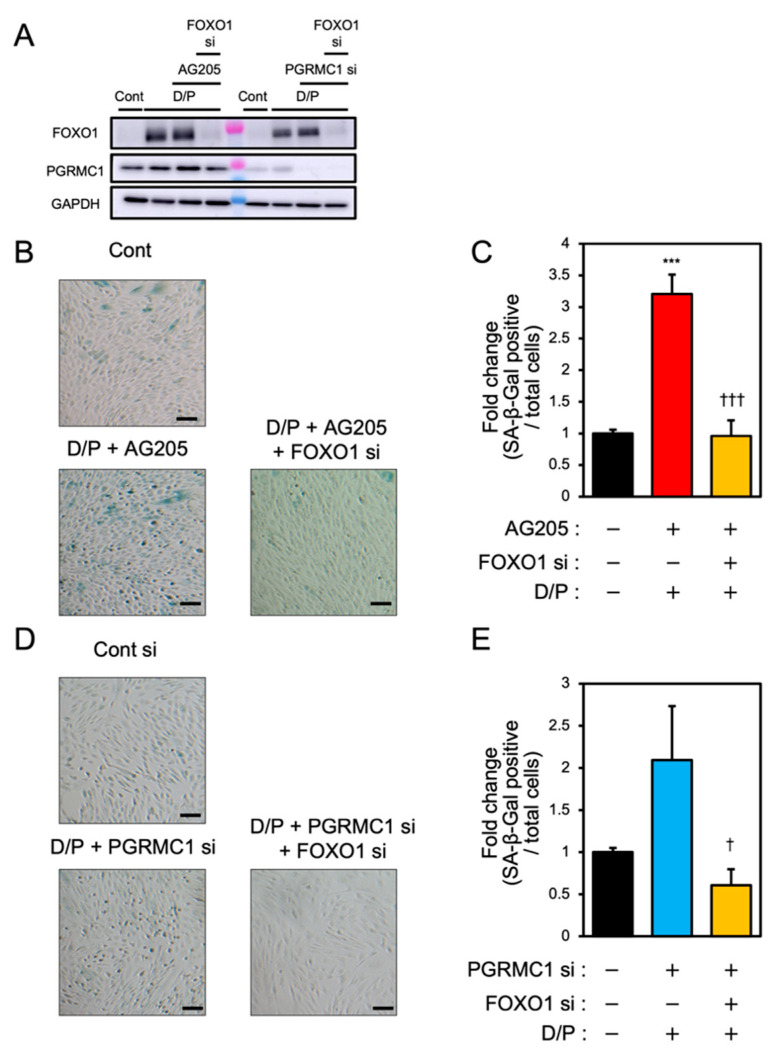
FOXO1 silencing ameliorates PGRMC1 inhibition- and knockdown-induced decidual senescence. ESCs transfected for 1 day with FOXO1 siRNA were treated with D/P in combination with AG-205 or PGRMC1 siRNA for 2 days. (**A**) Knockdown of PGRMC1 and FOXO1 was confirmed by immunoblotting. (**B**–**D**) Cellular senescence was analyzed by SA-β-Gal staining. (**C**,**E**) The graph shows the relative ratio of SA-β-Gal-positive cells to total cells. Values represent mean ± SEM from three independent experiments. *** *p* < 0.001 vs. control. † *p* < 0.05 vs. D/P + PGRMC1 siRNA. ††† *p* < 0.001 vs. D/P + AG-205. Scale bars = 100 μm. Red bar: AG205-treated group. Blue bar: the PGRMC1 siRNA-transfected group. Yellow bars: the PGRMC1/FOXO1 siRNA-transfected group.

**Figure 6 biomolecules-12-01046-f006:**
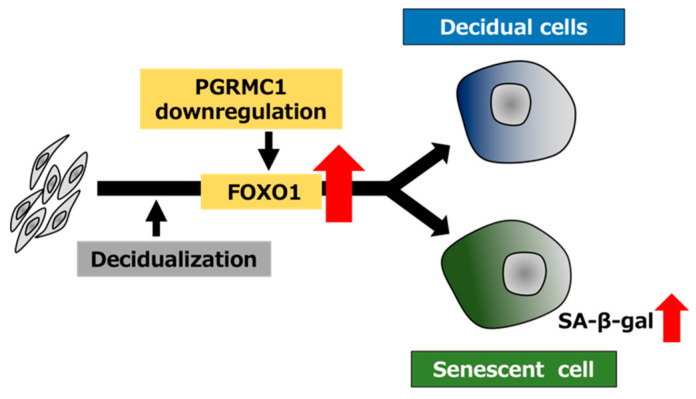
Schematic summary of PGRMC1 downregulation-mediated cellular senescence in decidualizing ESCs. The decrease in the PGRMC1 level in the secretory phase of the menstrual cycle may promote decidualization and concomitantly induce cellular senescence by upregulating FOXO1 expression in ESCs.

## Data Availability

Not applicable.

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
