# Peer review of "PGRMC1 Regulates Cellular Senescence via Modulating FOXO1 Expression in Decidualizing Endometrial Stromal Cells"

_biomolecules, 2022, doi:10.3390/biom12081046_

Round 1

Reviewer 1 Report

Tsuru and co-authors studied mechanisms underlying decidualization of human endometrial stromal cells in preparation for embryo implantation. They found that PGRMC1 is involved in cellular senescence during decidualization, which is a part of the natural process for acquisition of endometrial receptivity. Although the results are interesting, they are somewhat limited, likely because the results presented in this paper is only part of the study, and the other part is submitted elsewhere. Also, there are multiple concerns regarding the data presented in this paper.

Methods

2.2 Line 96: D/P is not defined

2.3 siRNA transfection was performed using lipofectamine. What was the efficiency of transfections? As these are primary cells, the transfection is usually performed by nucleofection.

2.4 Real time PCR: how many wells were used for one reaction?

2.6. Line 135 and figure legends state “The number of SA-β-gal positive cells…are expressed as a ratio compared with control group.” However, the graphs depict fold change from control group.

Results

3.2. Why was only IL8 measured by ELISA while other cytokines were not? It would be especially important in view of the discrepancy between IL8 mRNA and protein levels demonstrated in Results. Also, other SASP proteins including CXCL1 or IGFBPs could be included as markers of senescence. Western blots showing changes in p53/p21 and p16 would also confirm senescent phenotype.

3.3. FOXO1 is an important part of the pathway underlying endometrial stromal cell senescence. The authors argue that suppression of PGRMC1 enhances D/P induced-FOXO1 expression. This conclusion is based on the results of RT-PCR, in which FOXO1 increase is very incremental. I am not sure that a 30% increase in mRNA level will manifest a significant FOX1 protein  increase. Crucial demonstration by Western blot was not performed to confirm these results. Changes in cells after FOXO1 overexpression would additionally confirm the involvement of FOX1 in senescence in these experimental models.

3.4 Graph depicting changes in SA-β-gal after FOXO1 suppression is not provided.

3.5. Fig.5A: In this experiment, the effects of AG205 on PGRMC1 is not obvious,  and the bands of cells treated with siRNA are extremely weak. Also, in the legend, graphs are depicted in C and E, not C and D.

Author Response

Responses to Reviewer1’s comments

Tsuru and co-authors studied mechanisms underlying decidualization of human endometrial stromal cells in preparation for embryo implantation. They found that PGRMC1 is involved in cellular senescence during decidualization, which is a part of the natural process for acquisition of endometrial receptivity. Although the results are interesting, they are somewhat limited, likely because the results presented in this paper is only part of the study, and the other part is submitted elsewhere. Also, there are multiple concerns regarding the data presented in this paper.

Response: We appreciate your review of our manuscript, and constructive suggestions. Our responses to your comments are as follows.

Reviewer1

  1. Line 96: D/P is not defined

Response: The definition of D/P has been inserted into the sentence (lines 102-103 and 167).

  1. siRNA transfection was performed using lipofectamine. What was the efficiency of transfections? As these are primary cells, the transfection is usually performed by nucleofection.

Response: We understand your concern about the transfection. We did not calculate the actual transfection efficiency, but transfection of the PGRMC1 siRNA using lipofectamine effectively reduced PGRMC1 expression as shown in Figure 1C, D. Approximately 90% of PGRMC1 mRNA and protein expression was decreased by the siRNA treatment.

  1. Real time PCR: how many wells were used for one reaction?

Response: We conducted the real-time PCR using with double wells per each sample, and the series of experiment have been repeated three times for evaluation using primary ESCs isolated from three different patients.

  1. Line 135 and figure legends state “The number of SA-β-gal positive cells…are expressed as a ratio compared with control group.” However, the graphs depict fold change from control group.

Response: The term ‘The number of SA-β-gal positive cells…’ has been corrected to ‘Fold change’ throughout the manuscript (line 142-144 and figure legends).

  1. Why was only IL8 measured by ELISA while other cytokines were not? It would be especially important in view of the discrepancy between IL8 mRNA and protein levels demonstrated in Results. Also, other SASP proteins including CXCL1 or IGFBPs could be included as markers of senescence. Western blots showing changes in p53/p21 and p16 would also confirm senescent phenotype.

Response: According to this experiment, we measured the level of IL6 in the culture medium by ELISA (Figure 2D), and the results have been inserted in the section (lines 200-203), as described in the following.

“IL6 expression was suppressed in D/P-stimulated ESCs (Fig. 2C). AG205 increased mRNA expression and secretion of IL6 in the presence of D/P. However, PGRMC1 knockdown did not alter IL6 mRNA expression and the secretion under the D/P stimulation (Fig. 2C, D).”

As for other SASP factor IGFBP7, our preliminary data showed that the expression was increased by decidualization stimuli, and the treatment of AG205 promoted the D/P-stimulated IGFBP7 expression (Data not shown). Furthermore, the expression of p21 and p53 did not change after AG205 treatment and PGRMC1 knockdown in ESCs (Data not shown). These results suggest that enhanced cellular senescence by PGRMC1 inhibition and knockdown may not be mediated by p53 and p21 induction in ESCs. These sentences have been added to the discussion (lines 302-307).

  1. FOXO1 is an important part of the pathway underlying endometrial stromal cell senescence. The authors argue that suppression of PGRMC1 enhances D/P induced-FOXO1 expression. This conclusion is based on the results of RT-PCR, in which FOXO1 increase is very incremental. I am not sure that a 30% increase in mRNA level will manifest a significant FOX1 protein increase. Crucial demonstration by Western blot was not performed to confirm these results. Changes in cells after FOXO1 overexpression would additionally confirm the involvement of FOX1 in senescence in these experimental models.

Response: In addition to the data on FOXO1 mRNA expression, we confirmed that treatment with AG205 and PGRMC1 siRNA increased FOXO1 protein levels in the presence of D/P (Fig 5A).

As mentioned in introduction and result sections, FOXO1 is a critical inducer of cell senescence in ESCs during decidualization [6]. Because PGRMC1 inhibition and knockdown promoted FOXO1 expression in the presence of D/P, we examined the effect of FOXO1 knockdown, but not overexpression, on PGRMC1 inhibition and knockdown-induced senescence in ESCs.

  1. Graph depicting changes in SA-β-gal after FOXO1 suppression is not provided.

Response: According to your suggestion, we have added the graph in figure 4C. FOXO1 knockdown suppressed SA-β-Gal staining accompanying decidualization in ESCs (line 229-230).

  1. Fig.5A: In this experiment, the effects of AG205 on PGRMC1 is not obvious, and the bands of cells treated with siRNA are extremely weak.

Response: AG205 is an inhibitor of PGRMC1 that blocks the function of PGRMC1 protein [29 and 30], but not expression, by inhibiting its dimerization. Therefore, AG205 presumably does not alter the PGRMC1 expression. The description has been inserted in line 166.

  1. In the legend, graphs are depicted in C and E, not C and D.

Response: We have corrected the depict we had carelessly mistaken (line 249).

Reviewer 2 Report

The manuscript entitled as “PGRMC1 regulates cellular senescence via modulating FOXO1 expression in decidualizing endometrial stromal cells” is very interesting. However, I highly recommend the manuscript to go through revision and address the following questions before publication:

1.     How authors have optimized the concentration of PGRMC1 inhibitor, AG-205?

2.     I encourage to include the scale bar in the representative images of the cells.

3.     I suggest authors to perform the cell cycle analysis to evaluate in which cell cycle phase most of the cells are arrested.

4.     How authors have decided P4 dosage and time for its treatment to induce senescence?

5.     Did authors check if removal of P4 (1 μM) (D/P) treatment after 2 days can reverse the senescence induction?

6.     What could be the rationale, why PGRMC1 knockdown could not alter IL6 mRNA expression even after D/P stimulation, did they also evaluate the secretory level of IL6 in the cell supernatant?

7.     I recommend plotting the graph to show the relative ratio of SA-β-Gal-positive cells upon FOXO1 silencing in figure 4.

8.     I suggest immunoblot analysis to evaluate the potential of AG-205 inhibiting PGRMC1. Looking at the immunoblotting presented in Figure 5 A, it seems like the chosen concentration of AG-205 is not inhibiting PGRMC1.

9.     I also encourage to evaluate the mechanism of action in another cell line too.

  1. I strongly recommend improving the introduction and discussion section of the manuscript. Write how senescence could be an alternative therapeutic strategy against cancer. The below mentioned papers are suitable for citation:

Feng et al., 2019. Reprod Sci; 26(3): 394-403.

Cantonero et al., 2020. Int J Mol Sci; 21(20): 7641.

Dey et al., 2021. Mech Ageing Dev; 196:111497.

11.  I encourage to mention briefly in the discussion what P4 (1 μM) (D/P) treatment does? How does it induce senescence?

Author Response

Responses to Reviewer2’s comments

The manuscript entitled as “PGRMC1 regulates cellular senescence via modulating FOXO1 expression in decidualizing endometrial stromal cells” is very interesting. However, I highly recommend the manuscript to go through revision and address the following questions before publication:

Response: We appreciate your review of our manuscript, and constructive suggestions, our responses to your comments are as follows.

  1. How authors have optimized the concentration of PGRMC1 inhibitor, AG-205?

Response: Ten μM of AG205 is widely used to inhibit PGRMC1 action [31]. Our previous study confirmed that 10 μM of AG205 blocked PGRMC1 and stimulate the decidualization of endometrial stromal cells [27]. The description has been inserted in line 139.

  1. I encourage to include the scale bar in the representative images of the cells.

Response: Based upon your suggestion, we have added the scale bar in all pictures (Fig 1A, 1E, 4B, 5B, and 5D).

  1. I suggest authors to perform the cell cycle analysis to evaluate in which cell cycle phase most of the cells are arrested.

Response: According to your comments, the effects of PGRMC1 inhibition and knockdown on cell cycle progression were examined by FACS (Fig 1G). Cell cycle analysis showed that treatment with AG205 or PGRMC1 siRNA increased the population of the G2/M phase. (line 146-152, 175-176)

  1. How authors have decided P4 dosage and time for its treatment to induce senescence?

Response: We and others have confirmed that treatment of ESCs for 2 days with P4 (1 mM) and cAMP analog (0.5 mM) induced decidualization, accompanying with increased IGFBP-1 and prolactin [6, 27]. Because cellular senescence occurs with decidualization, we used the above dose of P4 and cAMP analog in this study.

  1. Did authors check if removal of P4 (1 μM) (D/P) treatment after 2 days can reverse the senescence induction?

Response: According to your comment, we examined the effects of the D/P withdrawal experiments. ESCs were treated with D/P for 2 days and then cultured for additional 2 days with or without the decidual stimuli. As shown attached Word file, the intensity of SA-b-Gal staining was similar to the D/P withdrawal ESCs, compared with that in ESCs stimulated for 4 days with D/P. It has been reported that the withdrawal of decidual stimuli resulted in de-differentiation with decreased prolactin and IGFBP1 expression (J Clin Endocrinol Metab. 2004; 89:5233). Moreover, our group has recently suggested that the removal of senescent cell by senolytic agents may stimulate decidualization [33]. However, this result here suggests that decidual senescence may not be reversible.

  1. What could be the rationale, why PGRMC1 knockdown could not alter IL6 mRNA expression even after D/P stimulation, did they also evaluate the secretory level of IL6 in the cell supernatant?

Response: According to your suggestion, the level of IL6 in the culture medium was measured by ELISA and the data has been added in Figure 2D. Treatment with AG205 increased IL6 secretion in the absence and presence of D/P. In contrast, PGRMC1 knockdown did not alter IL6 mRNA expression and secretion in D/P stimulation (Fig. 2C, D). These descriptions have been inserted in the result section (lines 200-203, 296-297).

  1. I recommend plotting the graph to show the relative ratio of SA-β-Gal-positive cells upon FOXO1 silencing in figure 4.

Response: We have added the graph in figure 4C. FOXO1 knockdown suppressed SA-β-Gal staining accompanying decidualization in ESCs(lines 229-230).

  1. I suggest immunoblot analysis to evaluate the potential of AG-205 inhibiting PGRMC1. Looking at the immunoblotting presented in Figure 5 A, it seems like the chosen concentration of AG-205 is not inhibiting PGRMC1.

Response: AG205 is an inhibitor of PGRMC1 that blocks the function of PGRMC1 [29, 30], but not expression, by inhibiting its dimerization. Therefore, AG205 presumably does not alter the PGRMC1 expression (line 166).

  1. I also encourage to evaluate the mechanism of action in another cell line too.

Response: In this study, we evaluated the effects of PGRMC1 inhibition and knockdown on decidual senescence using three different primary ESCs. We believe that primary ESCs are more suitable for examining the effects of functional inhibition of PGRMC1 on decidual senescence than an immortalized cell line that is artificially transduced hTERT or SV40.

  1. I strongly recommend improving the introduction and discussion section of the manuscript. Write how senescence could be an alternative therapeutic strategy against cancer. The below mentioned papers are suitable for citation: Feng et al., 2019. Reprod Sci; 26(3): 394-403. Cantonero et al., 2020. Int J Mol Sci; 21(20): 7641. Dey et al., 2021. Mech Ageing Dev; 196:111497.

Response: Thank you for your suggestion. We have modified the introduction and discussion sections as follows (lines 43-44, 49-52, 73-75, 270-272, 293-295).

Line 43: In general, DNA damage results in cellular senescence, which causes irreversible cell cycle arrest [10,11]. (Dey et al., 2021)

Line 73: In triple-negative breast cancer cells, the negative effect of P4 on cell proliferation and cell death was repressed by PGRMC1 silencing [25]. (Cantonero et al., 2020)

Line 293: Hydrogen peroxide-triggered senescence is further promoted by knockdown of PGRMC1 in fetal membrane cells [35], suggesting that PGRMC1 protects against oxidative stress-induced senescence in the fetal membrane. (Feng et al., 2019)

Lines 49: In cancer cells, SASP stimulate tumor growth and metastasis, but also trigger immune recognition and tumor clearance [12]. Although many questions currently remain un-answered concerning the cancer development which is associated with cellular senescence, senescence-based therapies warrant further investigation [13]. These sentences have been added to the introduction.

We have also added a description of the significance of senescent cell removal for decidualization instead of the relationship between senescence and cancer therapy as follows “Our group demonstrated that senescent cell removal by senolytic agents stimulates decidualization [33].”

  1. I encourage to mention briefly in the discussion what P4 (1 μM) (D/P) treatment does? How does it induce senescence?

Responese: As shown in figure 4 and other reports by Brighton et al., [6], FOXO1 expression is induced by D/P treatment and knockdown of the expression significantly repressed the senescence in ESCs. Although the precise mechanism of decidual senescence has not been well characterized.Brighton et al. has reported that FOXO1, which is upregulated upon decidualization, functions as a key inducer of cellular senescence via upregulating IL8 in ESCs. We have added the above description in discussion (lines 274-278).

Round 2

Reviewer 1 Report

The authors sucessfuly addressed my concerns

Reviewer 2 Report

I think the authors have improved the manuscript keeping the reviewer’s suggestions in mind. Therefore, I believe the manuscript can be accepted in the current form.